# Insight into the evolutionary assemblage of cranial kinesis from a Cretaceous bird

**Min Wang[1,2]\*, Thomas A Stidham[1,2,3], Jingmai K O'Connor[4], Zhonghe Zhou[1,2]**

[1]Key Laboratory of Vertebrate Evolution and Human Origins, Institute of Vertebrate Paleontology and Paleoanthropology, Chinese Academy of Sciences, Beijing, China; [2]Center for Excellence in Life and Paleoenvironment, Chinese Academy of Sciences, Beijing, China; [3]University of Chinese Academy of Sciences, Beijing, China; [4]Field Museum of Natural History, Chicago, United States

**Abstract** The independent movements and flexibility of various parts of the skull, called cranial kinesis, are an evolutionary innovation that is found in living vertebrates only in some squamates and crown birds and is considered to be a major factor underpinning much of the enormous phenotypic and ecological diversity of living birds, the most diverse group of extant amniotes. Compared to the postcranium, our understanding of the evolutionary assemblage of the characteristic modern bird skull has been hampered by sparse fossil records of early cranial materials, with competing hypotheses regarding the evolutionary development of cranial kinesis among early members of the avialans. Here, a detailed three-dimensional reconstruction of the skull of the Early Cretaceous enantiornithine *Yuanchuavis kompsosoura* allows for its in-depth description, including elements that are poorly known among early-diverging avialans but are central to deciphering the mosaic assembly of features required for modern avian cranial kinesis. Our reconstruction of the skull shows evolutionary and functional conservation of the temporal and palatal regions by retaining the ancestral theropod dinosaurian configuration within the skull of this otherwise derived and volant bird. Geometric morphometric analysis of the palatine suggests that loss of the jugal process represents the first step in the structural modifications of this element leading to the kinetic crown bird condition. The mixture of plesiomorphic temporal and palatal structures together with a derived avialan rostrum and postcranial skeleton encapsulated in *Yuanchuavis* manifests the key role of evolutionary mosaicism and experimentation in early bird diversification.

**\*For correspondence:**
wangmin@ivpp.ac.cn

**Competing interest:** The authors declare that no competing interests exist.

## Editor's evaluation

Most birds today can lift the upper beak independently of the brain case, enabled by a series of mobile joints and bending zones in the skull. The computed tomography of the skull of a 120-million-year-old toothed bird produced by the authors shows for the first time that the joints were still absent, but also hints at how they may have evolved later. This compelling, important paper is of high interest to evolutionary biologists, vertebrate paleontologists (especially, but by no means only, those working on bird origins) and specialists in biomechanics.

## Introduction

The growing fossil record of Mesozoic avialans has revealed that the initial appearance of many evolutionary novelties associated with living birds in fact originated among early lineages (*Xu et al., 2014*; *Brusatte et al., 2015*; *Chiappe and Meng, 2016*). Much of the research interest in the early evolution of birds has focused on flight-relevant morphologies (*Sullivan et al., 2017*; *Heers et al., 2021*). However, little is known about how the bulky and akinetic dinosaurian skull became transformed

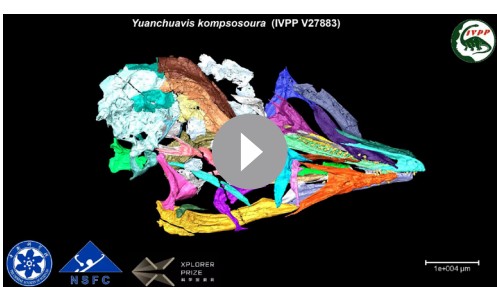

**Video 1.** Three-dimensional digital model of *Yuanchuavis kompsosoura* with rotation around vertical axis.

https://elifesciences.org/articles/81337/figures#video1

into the delicate, kinetic, and functionally diverse extant bird skull (*Bhullar et al., 2016*), which is inferred to have contributed significantly to their spectacular adaptive radiation, culminating in >10,000 species recognized today (*Zusi, 1984*; *Bout and Zweers, 2001*; *Gill, 2007*; *Bhullar et al., 2016*; *Olsen and Gremillet, 2017*). This lack of resolution results in part from the rarity of cranial materials of early avialans and the limited data available from conventional methods of examination given their typically poor and two-dimensional preservation (*Witmer and Martin, 1987*; *O'Connor and Chiappe, 2011a*). Meanwhile, some previous studies have perhaps over-stated the cranial resemblance between stem and crown birds, including positing the existence of certain bony contacts and intracranial joints (e.g. pterygoid-quadrate joint) required for kinesis, though yet to be confirmed positively in fossil taxa (*Witmer and Martin, 1987*; *Holliday and Witmer, 2008*; *Wang et al., 2021a*). As a step forward in our understanding of avialan cranial evolution, we applied x-ray CT scanning to the recently described Early Cretaceous bird *Yuanchuavis kompsosoura* (*Video 1* and *Video 2*; *Wang et al., 2021b*). *Yuanchuavis* phylogenetically is assigned to the second earliest branching enantiornithine clade (Pengornithidae) and preserves a unique pintail feather morphotype that has been inferred to be used for sexual display (*Wang et al., 2021b*). Our study documents previously unrecorded and unrecognized cranial features with clear functional and macroevolutionary implications.

## Results

Like in other pengornithids (*Zhou et al., 2008*; *Hu et al., 2015*; *O'Connor et al., 2016a*), the premaxillae are unfused. As in most Early Cretaceous enantiornithines including other pengornithids except for *Chiappeavis* (*Zhou et al., 2008*; *Hu et al., 2015*; *O'Connor et al., 2016a*), the short premaxillary corpus is rostrocaudally longer than dorsoventrally high, with subparallel dorsal and ventral margins (*Figures 1 and 2a*; *Figure 1—figure supplement 1*). The maxillary process of the premaxilla extends beyond the rostrocaudal midpoint of the external naris, which is proportionally longer than the state in other pengornithids where it terminates caudally within the rostral fourth of the external naris (*Zhou et al., 2008*; *Hu et al., 2015*; *O'Connor et al., 2016a*). Six premaxillary teeth are present on each side (*Figure 2b*), not five as estimated in a previous study (*Wang et al., 2021b*), and this is greater than the maximum number of four in other avialans (*Louchart and Viriot, 2011*; *O'Connor and Chiappe, 2011a*; *Hu et al., 2019*). As in other pengornithids (*Zhou et al., 2008*; *Hu et al., 2015*; *O'Connor et al., 2016a*), the tooth crowns are weakly expanded, and the tapered apices are slightly recurved. The frontal processes of the premaxillae become dorsoventrally compressed as they extend caudally, and the distal quarter of the process is caudolaterally tapered such that the two premaxillae define a medial notch for the nasals. The triradiate maxilla constitutes the major portion of the facial margin as in other early avialans (*Figure 2c*; *O'Connor et al., 2011b*; *Wang et al., 2021a*). The ascending process lacks the fenestra present in *Pengornis* (*Zhou et al., 2008*). The broad jugal process is sharply constricted along its caudal fourth, a unique feature otherwise unknown among early-diverging avialans (*O'Connor and Chiappe, 2011a*; *Rauhut et al., 2018*; *Kundrát et al., 2019*; *O'Connor et al., 2020*; *Wang et al., 2021a*), although it is likely that this portion of the maxilla is commonly broken and lost or covered

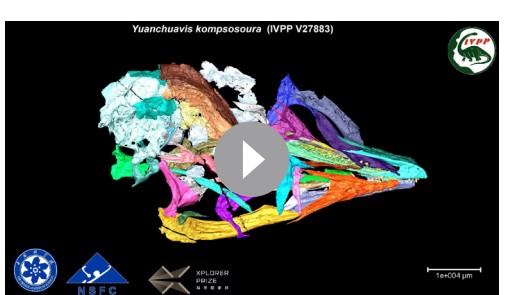

**Video 2.** Three-dimensional digital model of *Yuanchuavis kompsosoura* with rotation around horizontal axis.

https://elifesciences.org/articles/81337/figures#video2

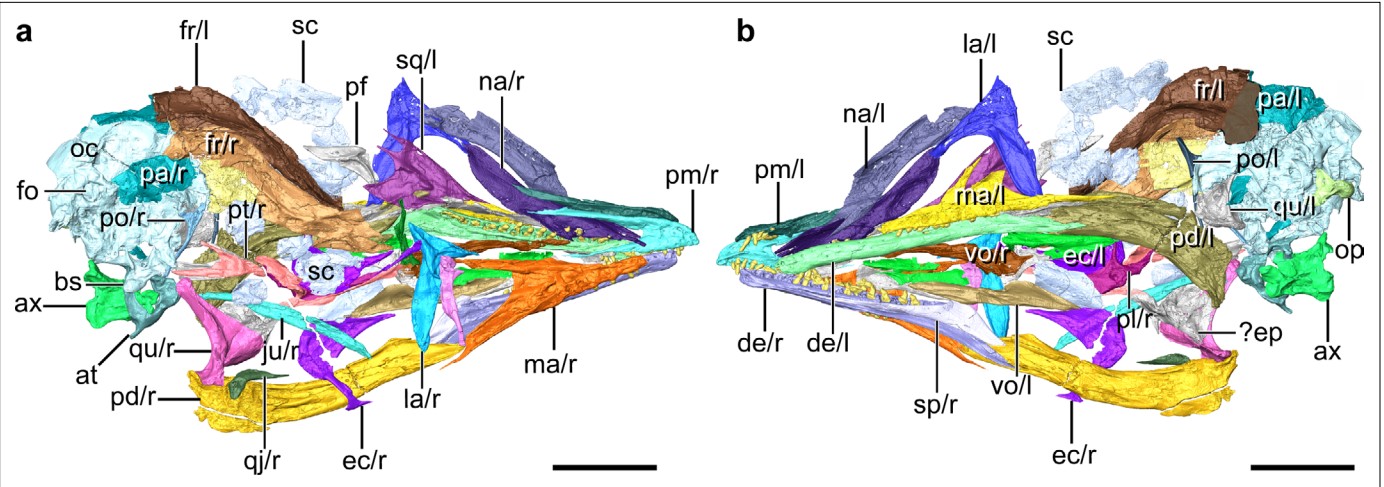

**Figure 1.** Digital reconstruction of the skull of *Yuanchuavis*, IVPP V27883. (**a and b**) Skull in right (**a**) and left (**b**) view, respectively. ax, axis; at, atlas; bs, basipterygoid process; de, dentary; ec, ectopterygoid; ep, epipterygoids; fo, foramen magnum; fr, frontal; ju, jugal; la, lacrimal; ma, maxilla; na, nasal; oc, occipital region; op, occipital condyle; pa, parietal; pd, post-dentary mandible; pl, palatine; pm, premaxilla; po, postorbital; pt, pterygoid; qj, quadratojugal; qu, quadrate; sc, scleral ossicles; sp, splenial; sq, squamosal; vo, vomer; and r/l, right/left side. Scale bars, 10 mm.

The online version of this article includes the following figure supplement(s) for figure 1:

**Figure supplement 1.** Cranial anatomy of *Yuanchuavis*, IVPP27883.

**Figure supplement 2.** Digital reconstruction of atlas and axis of *Yuanchuavis*.

by the jugal. The ventral margin of the jugal process flares laterally, forming a lateral groove that accommodates the jugal. Nine maxillary teeth are present (seven preserved teeth and two additional alveoli preserved in the right maxilla), which like in other pengornithids exceeds the number present in some enantiornithines (e.g. bohaiornithids; *O'Connor and Chiappe, 2011a*; *Wang et al., 2014*). The exact number of maxillary teeth is unclear in other pengornithids.

The nasal is weakly bowed ventrally and tapered at both ends (*Figure 2d*). A maxillary process, present in *Parapengornis* and other enantiornithines (*Zhou et al., 2005*; *O'Connor et al., 2011b*; *Wang et al., 2021b*), is absent. The elongate nasals contact medially for almost their entire length such that the premaxillae and frontals do not contact. The rostral end of the nasal is forked as in *Pengornis* (*Zhou et al., 2008*) and receives the caudal tapered portion of the nasal process of the premaxilla (*Figure 2—figure supplement 1a, b*). The nasal has a width-to-length ratio of 0.11, notably more slender than in some enantiornithines (e.g. *Eoenantiornis* and *Protopteryx*) but wider than in *Falcatakely* (*O'Connor et al., 2020*). In contrast to *Pengornis* (*Zhou et al., 2008*), the bone is not perforated by a foramen.

The lacrimal is triradiate with a robust descending ramus that is longer and wider than the more delicate rostral ramus, which in turn is twice the length of the caudal ramus (*Figure 2e*). The straight dorsal margin of the lacrimal lacks the central concavity characteristic of *Pengornis* (*Zhou et al., 2008*). The descending ramus widens rostrocaudally along its central portion and then narrows toward the ventral end, lacking the ventral expansion seen in *Falcatakely* and *Ichthyornis* (*O'Connor et al., 2020*; *Torres et al., 2021*). An oval fenestra is developed near its caudal margin at the dorsoventral midpoint (recognized on both sides), which is absent in other early avialans including other pengornithids (*Mayr et al., 2005*; *O'Connor and Chiappe, 2011a*; *Zhou et al., 2013*; *O'Connor et al., 2020*). The lacrimal bears a triradiate lateral embossment forming crests along each ramus, and it is centered at the juncture of the three rami. No similar structure has yet been described among other early avialans (*O'Connor et al., 2011b*; *Rauhut et al., 2018*; *Wang et al., 2021a*), but this morphology is present in deinonychosaurians (*Norell et al., 2006*; *Xu et al., 2015*). While the dorsal branches of the lateral embossment are relatively straight, the ventral branch curves with a concave rostral margin, and it decreases in height toward its tip, ending in the dorsal one-fifth of the rostral margin of the descending ramus. The preserved left prefrontal is similar to the condition in *Archaeopteryx* (*Rauhut et al., 2018*) and consistent with the relatively short proportion of the caudal ramus of the lacrimal (*Figure 1*).

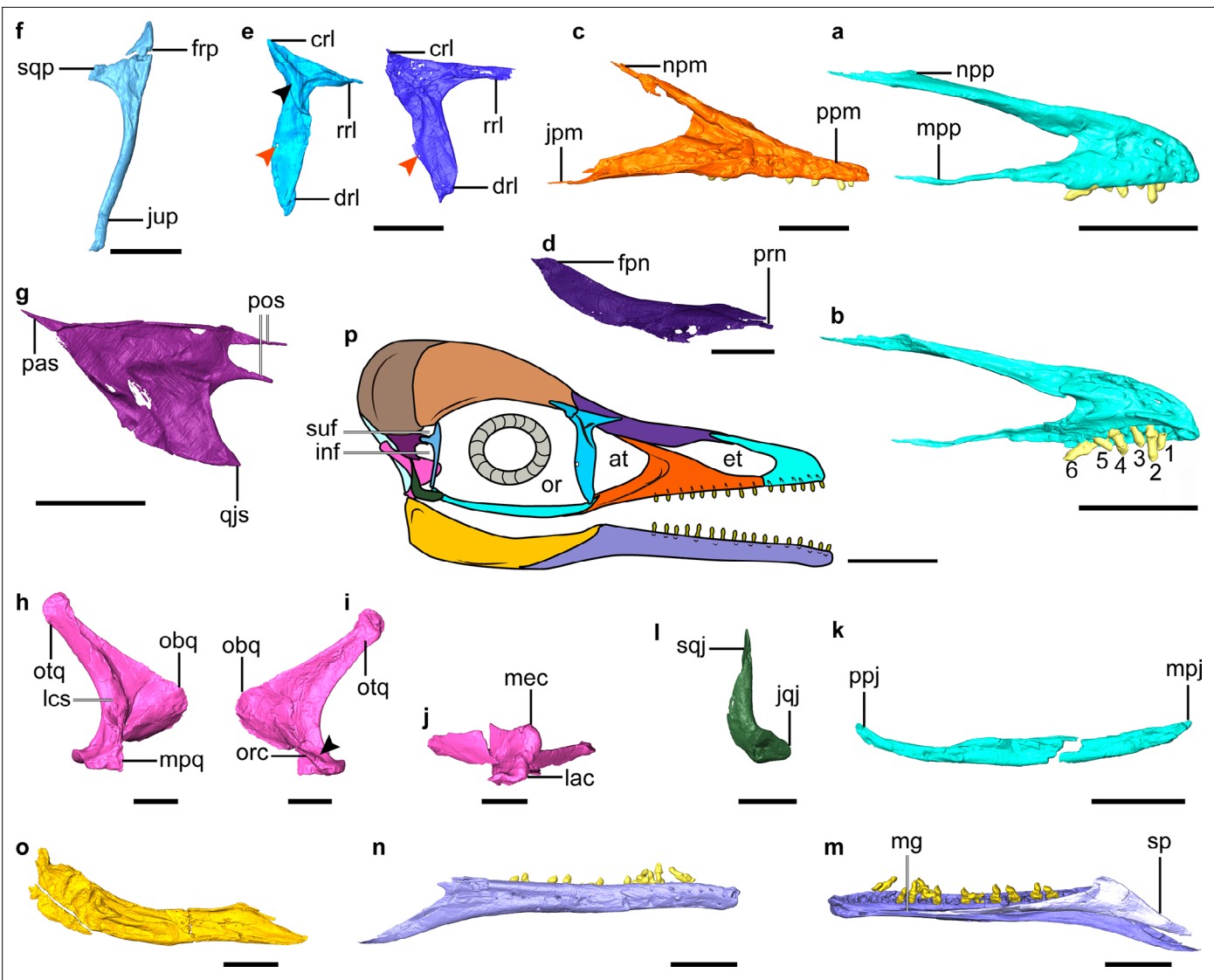

**Figure 2.** Digital reconstruction of facial and jaw bones of *Yuanchuavis*. (**a and b**) Right premaxilla in lateral and medial view, respectively (numbers 1–6 denote premaxillary teeth; (**b**) is mirrored). (**c**) Right maxilla. (**d**) Right nasal. (**e**) Right (right column) and left lacrimal in lateral and medial view, respectively (black and red arrowhead denote the lateral flange and foramen, respectively). (**f**) Right postorbital. (**g**) Left squamosal. (**h–j**) Right quadrate (**i**) arrowhead denotes the absence of a pterygoid condyle. (**k**) Right jugal. (**l**) Right quadratojugal. (**m and n**) Right dentary. (**o**) Right post-dentary mandible. (**p**) Reconstruction of *Yuanchuavis* skull in lateral aspect. (**a, c–f, h, k–m,** and **o**) Lateral view; (**b, g, i,** and **n**) medial view; (**j**) ventral view. at, antorbital fenestra; crl, caudal ramus of lacrimal; drl, descending ramus of lacrimal; et, external naris; frp, frontal process of postorbital; inf, infratemporal fenestra; jpm, jugal process of maxilla; jqj, jugal process of quadratojugal; jup, jugal process of postorbital; lac, lateral condyle; lsc, lateral crest; mec, medial condyle; mg, Meckel's groove; mpj, maxillary process of jugal; mpp, maxillary process of premaxilla; mpq, mandiblar process of quadrate; npm, nasal process of maxilla; npp, nasal process of premaxilla; obq, orbital process of quadrate; or, orbit; orc, orbitocondylar crest; otq, otic process of quadrate; pas, paroccipital process of squamosal; pf, prefrontal; pos, postorbital process of squamosal; ppj, postorbital process of jugal; ppm, premaxillary process of maxilla; qjs, quadratojugal process of squamosal; rrl, rostral ramus of lacrimal; sp, splenial; sqj, squamosal process of quadratojugal; sqp, squamosal process of postorbital; suf, supratemporal fenestra. Scale bars, 5 mm (**a–e, g–k,** and **m–o**), 2.5 mm (**f and l**), and 10 mm (**p**).

The online version of this article includes the following figure supplement(s) for figure 2:

**Figure supplement 1.** Additional cranial anatomy of *Yuanchuavis*.

---

The postorbital is an enigmatic element poorly known among enantiornithines (likely independently reduced in some taxa; *Xu et al., 2021*) but is preserved exquisitely here (*Figure 2f*). The Y-shaped postorbital has a pointed frontal process and a blunter squamosal process that is oriented at a near right angle to one another. The dorsal margin between the frontal and squamosal processes is concave,

forming the rostroventral margin of the supratemporal fenestra. The squamosal process is shorter than the frontal process and ends in a blunt articular surface for the squamosal. The jugal process is approximately four times longer than the frontal process, delicate, and deflected caudoventrally. The squamosal is another poorly known cranial element in enantiornithines that has been recognized in only two specimens (LP-4450-IEI and IVPP V12707; *O'Connor and Chiappe, 2011a*). Compared to those specimens, the squamosal of *Yuanchuavis* is morphologically more similar to that of *Archaeopteryx* and non-avialan theropods such as *Deinonychus*, attesting to its relatively large size, the robust quadratojugal process, and the rostrally deeply forked postorbital process (*Figure 2g*). In contrast, the postorbital process is not forked in LP-4450-IEI and IVPP V12707 (*O'Connor and Chiappe, 2011a*; *Wang et al., 2021a*). The notch in the postorbital process is deeper than in *Archaeopteryx* (*Elżanowski and Wellnhofer, 1996*) but not as pronounced as in *Deinonychus* (*Ostrom, 1969*). As in *Archaeopteryx* (*Elżanowski and Wellnhofer, 1996*), the dorsal and ventral branches of the postorbital process are subequal in length; whereas the ventral branch is longer in *Deinonychus* (*Ostrom, 1969*). The trapezoidal-shaped quadratojugal process bears a pointed rostroventral corner that projects as far rostrally as the postorbital process. In contrast, the quadratojugal process is slender and rod-like in *Archaeopteryx* and other enantiornithines and is triangular in non-avialan theropods (*Ostrom, 1969*; *Norell et al., 2006*; *O'Connor et al., 2011b*; *Xu et al., 2015*; *Wang et al., 2021a*). The paroccipital process is sharply tapered as it projects caudally, and it is connected with the quadratojugal process by a roughly straight ventral margin, resembling the condition in *Linheraptor* (*Xu et al., 2015*). By contrast, that process is caudoventrally directed, resulting in a right angle with the quadratojugal process in IVPP V12707 and *Deinonychus* (*Ostrom, 1969*; *Wang et al., 2021a*).

The unfused frontals and parietals are crushed, revealing few features except that the skull roof is vaulted dorsally. There is an elongate ridge along the lateral edge of the frontal forming a concave margin around the dorsal margin of the orbit, and this ridge runs for most of the length of the frontal. The frontal does not appear to have contributed to a postorbital process.

As in other early avialans and non-avialan theropods (*Turner et al., 2012*; *Hendrickx et al., 2015*; *Wang et al., 2021a*), the quadrate has a bicondylar mandibular process (*Figure 2h–j*, *Figure 2—figure supplement 1c-e*). Like in some other enantiornithines but not *Longipteryx* (*O'Connor and Chiappe, 2011a*; *Stidham and O'Connor, 2021*), the medial condyle is larger than the lateral one (*Figure 2j*). The caudal surface of the quadrate corpus is not perforated by a foramen as in some enantiornithines including *Pengornis* and *Shenqiornis* (*Zhou et al., 2008*; *O'Connor and Chiappe, 2011a*). In lateral view, the caudal margin of the quadrate is concave and bowed with the medial and lateral mandibular condyles positioned caudally relative to the corpus as in the juvenile enantiornithine IVPP V12707 (*Wang et al., 2021a*), rather than being level with the shaft as in non-avialan theropods such as *Linheraptor* (*Turner et al., 2012*; *Xu et al., 2015*). The bowing is restricted to the ventral half of the quadrate with the caudal margin (lateral view) being relatively straight in the dorsal half, and the absence of cracks in the corpus supports this as the original morphology (*Figure 2h and j*). This morphology also suggests that the quadrate was inclined with the otic process positioned caudal to the mandibular process. There is a shallow concavity separating the medial and lateral condyles near the mediolateral midpoint of the mandibular process (*Figure 2—figure supplement 1c*). There does not appear to be a rostral or caudal labrum around the condyles, but the medial condyle faces a bit caudally and is raised up above the adjacent bone surface (also visible on the left quadrate). The lateral edge of the mandibular process is broken so that the morphology of the articulation with the quadratojugal cannot be identified in detail, but the remnants are directed and extended somewhat rostrally, in a manner similar to the state present in *Linheraptor* where it has rostrocaudally elongate contact with the quadratojugal. That elongation has not been identified in other enantiornithines. This rostrally directed process is ventral to the ventral edge of the orbital process. The caudal surface of the quadrate corpus forms a sharp crest along its dorsoventral length from the otic head ventrally to just dorsal to the concavity between the medial and lateral mandibular condyles. That sharp caudal crest is present in many, but not all enantiornithines. Like IVPP V12707 and non-avialan theropods (*Norell et al., 2006*; *Hendrickx et al., 2015*; *Wang et al., 2021a*), a distinct pterygoid condyle is absent on the quadrate (*Figure 2—figure supplement 1c*), indicating the absence of a condylar-based joint between the quadrate and pterygoid, a derived feature present in *Ichthyornis*, hesperornithiforms, and crown taxa (*Gingerich, 1976*; *Baumel and Witmer, 1993*; *Field et al., 2018b*). The shape of the orbital process is nearly identical to that of *Archaeopteryx* and non-avialan theropods (called the

pterygoid ramus) in having a rounded and blunt convex rostral apex with a straight dorsal margin that extends to the otic process (*Figure 2h*; *Currie, 1995*; *Norell et al., 2006*; *Hendrickx et al., 2015*; *Rauhut et al., 2018*). In contrast, the orbital process is dorsoventrally thin and tapers rostrally in other enantiornithines (*Sander et al., 2001*; *Stidham and O'Connor, 2021*; *Wang et al., 2021a*), which is further modified into a narrower, pointed process in *Ichthyornis* and more crownward taxa (*Elżanowski and Stidham, 2011*; *Field et al., 2018b*; *Torres et al., 2021*). In addition, its apex is in the ventral half of the quadrate like *Sapeornis* and enantiornithines but differs from the position near the dorsoventral midpoint in many non-avialans, *Archaeopteryx*, and ornithurines (*Elżanowski and Stidham, 2011*; *Field et al., 2018b*; *Stidham and O'Connor, 2021*; *Wang et al., 2021a*). The orbital process extends dorsally, and it lowers in height ending ventral to the undivided otic head. The lateral surface of the orbital process is relatively smooth. The medial side appears a bit rougher, and its dorsal edge thickens rostrally (*Figure 2—figure supplement 1c*). The ventral margin of the orbital process thickens ventrally near its base and extends ventrally as an orbitocondylar crest, contacting the area near the dorsal edge of the medial apex of the medial mandibular condyle (*Figure 2i*, *Figure 2—figure supplement 1a*). The orbitocondylar crest is not yet known among other enantiornithines (*Elżanowski and Stidham, 2011*; *Field et al., 2018b*; *Wang et al., 2021a*). As in other enantiornithines and most non-avialans (*Stidham and O'Connor, 2021*; *Wang et al., 2021a*), the otic head is not divided into separate squamosal and otic capitula. The otic process has a triangular cross section, with the caudal crest on the corpus flanked by flattened medial and lateral surfaces, as in other enantiornithines. As in some enantiornithines such as *Longipteryx* (*Stidham and O'Connor, 2021*), a separate lateral crest extends from the otic process to the lateral edge of the mandibular process (*Figure 2h*) and bounds a caudolateral fossa on the ventral caudal surface of the quadrate corpus. The crest has been crushed along much of its length, but it appears to have extended dorsally closer to the otic head than the orbital process. There is no evidence of a distinct fossa dorsal to the rostral face of the mandibular condyles, although the area near the rostrally projected apex of the lateral mandibular process is concave. It is unclear if there was an orbital fossa because of folding distortion of the orbital process, but the area dorsal to the orbitocondylar crest is somewhat concave. There is no evidence of a pneumatic foramen in that region or anywhere on the bone.

A fragment preserved and compressed to the medial surface of the orbital process of the right quadrate is tentatively identified as the epipterygoid (*Figure 1*), given its proximity with the right pterygoid and current overlapping state with the quadrate. No epipterygoid has been reported previously among early-diverging avialans, but it is known in some avialan outgroups.

As in *Pterygornis* (*Wang and Hu, 2017*), the caudal end of the jugal curves caudodorsally rather than being forked as in other pengornithids and enantiornithines such as *Falcatakely* and IVPP V12707 (*Figure 2k*; *O'Connor and Chiappe, 2011a*; *Wang et al., 2021a*). The L-shaped quadratojugal has a dorsally directed postorbital process (with a groove on its caudal dorsal edge presumably for articulation with the squamosal) and a short jugal process (*Figure 2l*).

The pterygoid has a large, forked, and caudodorsally directed quadrate ramus (*Figure 3a, b*), and the pterygoid does not contact the quadrate on its ventromedial aspect through a condyle as in ornithurines (*Baumel and Witmer, 1993*). The occurrence of this morphology in *Yuanchuavis* reinforces the recent identification of the retention of this plesiomorphic curved and forked non-avialan pterygoid shape among enantiornithines (*Figure 3—figure supplement 1*; *Wang et al., 2021a*). The mediolaterally compressed quadrate ramus is forked into longer dorsal and shorter ventral processes, opposite of the condition seen in the enantiornithine IVPP V12707 and dromaeosaurids (*Figure 3—figure supplement 1a–d*; *Ostrom, 1969*; *Barsbold and Osmólska, 1999*; *Tsuihiji et al., 2014*; *Wang et al., 2021a*). By contrast, the ornithurine pterygoid is modified substantially with a reduced or absent quadrate ramus and an overall elongation reaching to the area just dorsal to the medial mandibular condyle of the quadrate, as seen in the triangular shape in hesperornithiforms and overall more strut-like appearance in crown birds (*Figure 3—figure supplement 1e*; *Gingerich, 1976*; *Baumel and Witmer, 1993*; *Zusi and Livezey, 2006*). A potentially homologous, though reduced, pterygoid dorsal process extends dorsally above the pterygoid condyle of the quadrate along the medial base of the orbital process into the orbital fossa in paleognaths like the ostrich and emu (*Figure 3—figure supplement 1f*; *McDowell, 1948*; *Zusi and Livezey, 2006*). The leaf-like palatine process is constricted along the middle third of the shaft and then expands mediolaterally rostrally. The caudal end of the palatine process extends caudally well beyond the level of the cotyle for the basipterygoid process, but

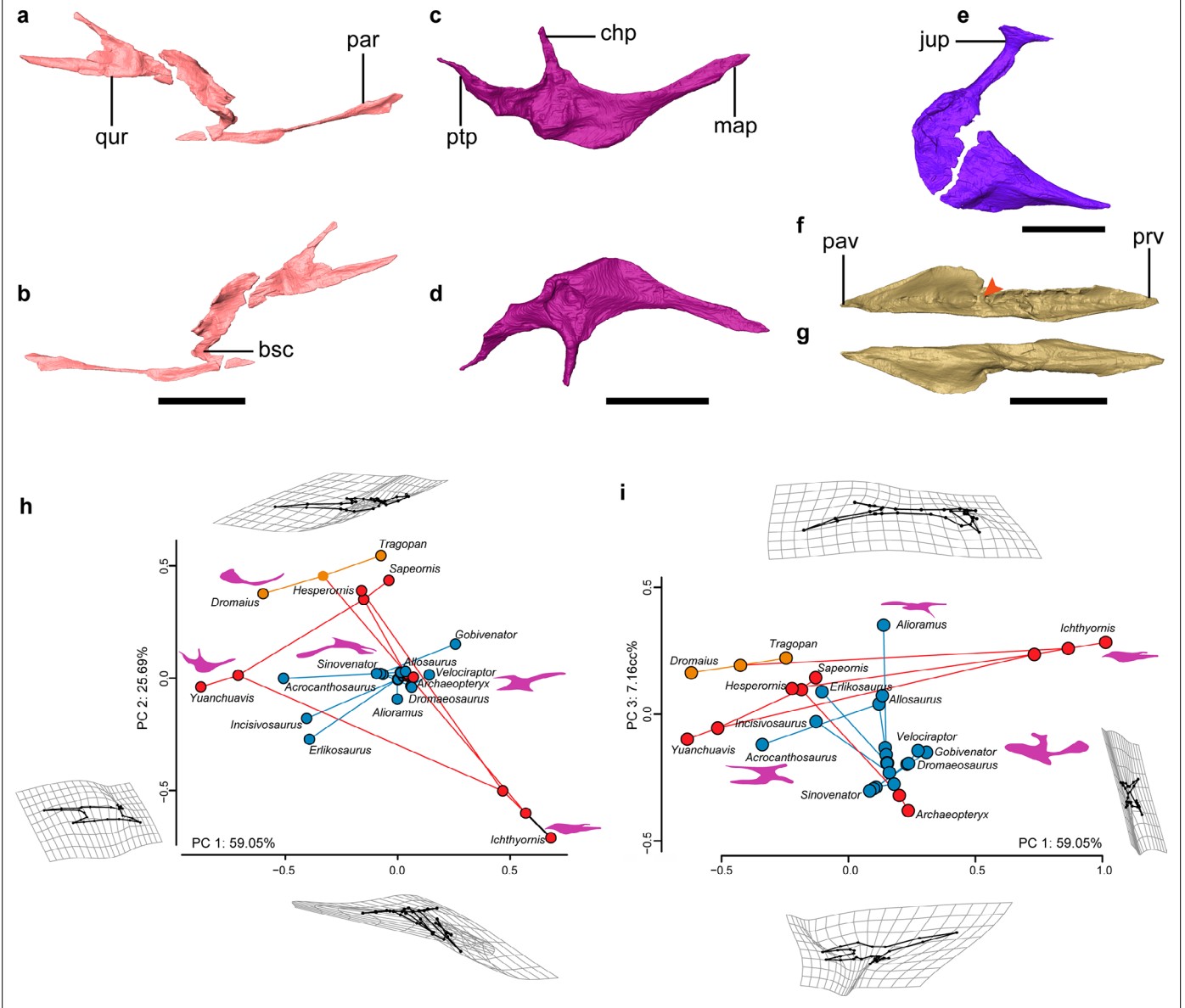

**Figure 3.** Palate anatomy of *Yuanchuavis*. (**a–f**) Digital reconstruction of the right pterygoid in lateral (**a**) and ventral (**b**) view; right palatine in dorsal (**c**) and ventral (**d**) view; right ectopterygoid in dorsal view (**e**); and left vomer in dorsal (**f**) and ventral (**g**) view (arrowhead denotes the dorsal transverse ridge). (**h and i**) Phylomorphospace showing the diversity of palatine shape in early-diverging avialans and their close non-avialan theropod relatives based on the first three principal components (PC1–PC3), with deformation grids and wireframes from average to extreme; line drawings of the palatine in dorsal/ventral view are placed nearby corresponding taxa (blue circles: non-avialan theropods; red circles: Mesozoic avialans; orange circles: crown birds). bsc, basipterygoid process cotyle; chp, choanal process; jup, jugal process; map, maxillary process; par, palatine ramus; pav, palatine ramus of vomer; prv, premaxillary ramus of vomer; ptp, pterygoid process; qur, quadrate ramus. Scale bars, 10 mm (**a–d, f, and g**), 2 mm (**e**).

The online version of this article includes the following figure supplement(s) for figure 3:

**Figure supplement 1.** Comparison of pterygoid morphology.

**Figure supplement 2.** Comparison of palatine morphology.

**Figure supplement 3.** Comparison of vomer morphology.

**Figure supplement 4.** Diversity of palatine shape in early-diverging avialans and non-avialan theropods.

**Figure supplement 5.** Landmark scheme.

the corresponding portion is reduced in IVPP V12707 and non-avialan theropods (*Figure 3—figure supplement 1*; *Ostrom, 1969*; *Barsbold and Osmólska, 1999*; *Wang et al., 2021a*).

While knowledge of the morphology of the palatine has been elusive among early avialans, the bone is well preserved in *Yuanchuavis* (*Figure 3c and d*). As in *Falcatakely* (*O'Connor et al., 2020*), the bone is triradiate and lacks a jugal process present in *Archaeopteryx* and non-avialan dinosaurs (*Figure 3—figure supplement 2a–f*; *Ostrom, 1969*; *Barsbold and Osmólska, 1999*; *Mayr et al., 2007*; *Kundrát et al., 2019*). A jugal process is not preserved in the early pygostylian *Sapeornis* (*Figure 3—figure supplement 2b*), which was interpreted as preservational bias (*Hu et al., 2019*). Given the shared condition and the exquisite preservation in *Yuanchuavis* and *Falcatakely*, we posit that the loss of the jugal process is genuine in these taxa and represents a derived condition other-wise widely distributed among crownward ornithuromorphs (*Figure 3—figure supplement 2c, g, h*; *Elżanowski, 1991*; *Elżanowski and Wellnhofer, 1996*; *Zusi and Livezey, 2006*). As in *Sapeornis* and *Hesperornis* (*Elżanowski, 1991*; *Hu et al., 2019*), the maxillary process is longer than the pterygoid process, in contrast to the condition in *Archaeopteryx* (*Figure 3—figure supplement 2*; *Elżanowski and Wellnhofer, 1996*). As in *Sapeornis* and *Falcatakely*, the caudally tapering pterygoid process is caudomedially directed with respect to the maxillary process, rather than being in line with the latter as in *Hesperornis* (*Elżanowski, 1991*; *Elżanowski and Wellnhofer, 1996*), and some non-avialan theropods (e.g. *Velociraptor*; *Barsbold and Osmólska, 1999*). As in *Sapeornis* (*Hu et al., 2019*), the choanal process projects medially, rather than being hooked rostrally as in other early avialans (*Elżanowski, 1991*; *Elżanowski and Wellnhofer, 1996*; *O'Connor et al., 2020*), and some non-avialan theropods (*Currie, 1995*; *Barsbold and Osmólska, 1999*).

The ectopterygoid has a hooked jugal process that is expanded rostrocaudally at the lateral end, forming a large contact facet for the jugal, as in *Sapeornis* and some non-avialan theropods like *Allosaurus* (*Figure 3e*; *Madsen, 1976*; *Hu et al., 2019*). In contrast, a similar expansion is absent among other early avialans such as *Archaeopteryx* and IVPP V12707 (*Elżanowski and Wellnhofer, 1996*; *Wang et al., 2021a*). The main body of the ectopterygoid is fan-shaped, and its rostral end projects more rostrally than the state in *Archaeopteryx* and *Sapeornis* (*Mayr et al., 2005*; *Hu et al., 2019*).

The vomers are separated from each other as in the juvenile enantiornithine IVPP V127071 (*Figure 3f and g*; *Figure 3—figure supplement 3a, b*), suggesting that the lack of fusion in the latter cannot be unequivocally attributed to ontogeny given that the holotype specimen here is from an osteologically mature individual. However, fused vomers are present in *Gobipteryx*, *Sapeornis*, and most crown birds (*Baumel and Witmer, 1993*; *Chiappe et al., 2001*; *Hu et al., 2019*; *Figure 3—figure supplement 2c*), and that variation demonstrates species-specific development of this element. The vomers differ from that of other known Mesozoic avialan and non-avialan theropods in that each vomer is concave dorsally with a thickened medial margin, and a dorsal transverse ridge is present at midshaft, dividing the vomer into premaxillary and pterygoid rami (*Figure 3f*, *Figure 3—figure supplement 2*; *Ostrom, 1969*; *Elżanowski and Wellnhofer, 1996*; *Lautenschlager et al., 2014*; *Hu et al., 2019*; *Wang et al., 2021a*). The premaxillary ramus is spear-shaped in dorsal view. The pterygoid ramus reaches its maximum mediolateral width at its midpoint and narrows caudally. Unlike the medio-laterally compressed form observed in IVPP V12707 (*Wang et al., 2021a*), the pterygoid ramus is dorsoventrally compressed and forms a dorsal groove to articulate with the pterygoid. A caudodorsal process, projecting from the caudal end of the vomer like that of *Sapeornis* and some non-avialan theropods (*Hu et al., 2019*), is absent.

The occipital elements around the foramen magnum are compressed, but the occipital condyle is distinct with a heart-shaped outline and a nerve foramen on the left side of its base. The blunt though pointed basipterygoid process is pronounced as in some enantiornithines (e.g. *Zhouornis* and *Brevirostruavis*; *Zhang et al., 2013*; *Wang et al., 2021b*), paleognaths (*Gussekloo and Bout, 2005*), and non-avialan theropods (*Holliday and Witmer, 2008*; *Figure 1*, *Figure 1—figure supplement 1*).

As in other enantiornithines (*O'Connor and Chiappe, 2011a*), the dentary has parallel dorsal and ventral margins and a caudoventrally sloping caudal margin (*Figure 2m, n*). 17 dentary teeth were present on each side (11 teeth in situ and 6 alveoli; *Figure 2—figure supplement 1f*). Meckel's groove extends rostrally to the level of the third dentary tooth and appears to be fully covered medially by the triangular splenial (*Figure 2—figure supplement 1f, g*). The post-dentary mandibular elements are compressed with each other (or fused partially), preventing digital segmentation of the individual elements (*Figure 2o*). Unlike IVPP V12707 (*Wang et al., 2021a*), neither a mandibular fenestra nor a

coronoid process is present. There appears to have been a small retroarticular process in *Yuanchuavis*, but a prominent one is present in some enantiornithines such as bohaiornithids and *Brevirostruavis* (*Wang et al., 2014*; *Li et al., 2021*). The caudal end of the right jaw is broken just caudal and ventral to its articulation with the quadrate, but the preserved base of the retroarticular process does extend somewhat dorsally. The caudal end of the left jaw appears possibly complete and exhibits an overall squared caudal end, suggesting the complete shape of the process.

The atlas has a complete fused neural arch, rendering this element ring-like in cranial view (*Figure 1—figure supplement 2*). The condyloid fossa is subcircular in outline and bears a shallow incisure fossa. A pair of short costal processes project caudodorsally as in some enantiornithines such as *Piscivorenantiornis* (*Wang and Zhou, 2017*). The articular facet for the axis is mediolaterally wider than dorsoventrally high. The elongate axis is craniocaudally longer than dorsoventrally high. The caudal articular facet of the centrum seems to be heterocoelic, being mediolaterally convex and dorsoventrally concave, as in crown birds.

## Discussion

The exquisitely preserved skull of *Yuanchuavis*, visualized through high-resolution x-ray CT, enables reconstruction of the cranial morphology of early avialans with great fidelity. In particular, this reconstruction includes elements that are otherwise rarely preserved or poorly known but are key to the analysis of macroevolutionary and functional properties of the skull (*Zusi, 1993*; *Sander et al., 2001*). Given the presence of pointed and lightweight jaw bones, enlarged orbit, and vaulted cranial roof, the skull is clearly bird-like. However, the morphologies of the temporal and palatal regions speak to their plesiomorphic nature and attest to the retention of typical non-avian dinosaurian conditions with little modifications. The combination of a free postorbital with an elongate jugal process and a large squamosal with a prominent postorbital process indicates the presence of postorbital and temporal bars that completely separate the supratemporal and infratemporal fenestrae and the orbit from each other (*Figure 2p*). In addition, the infratemporal fenestra likely is separated from the quadrate fenestra given the length of the squamosal process of the quadratojugal and the quadratojugal process of the squamosal. These observations unambiguously show that *Yuanchuavis* has a diapsid skull, which so far has been confirmed in *Archaeopteryx* (*Elżanowski and Wellnhofer, 1996*; *Mayr et al., 2005*), the early pygostylian *Sapeornis* (*Hu et al., 2019*), confuciusornithiforms (*Elżanowski et al., 2018*), and two enantiornithines (*Longusunguis* and IVPP V12707) among Cretaceous avialans (*Wang et al., 2021a*), showing that this ancestral temporal configuration is conserved evolutionarily well into the early avialan diversification; whereas the recently reported Late Cretaceous enantiornithine *Yuornis* shows the independent loss of the postorbital bar along with reduction of other cranial elements (*Xu et al., 2021*).

The structure of the palate is functionally vital to the feeding behaviors of vertebrates (*Zusi and Livezey, 2006*; *Holliday and Witmer, 2008*), and it underpins the diverse forms of cranial kinesis present in crown birds (*Zusi, 1984*; *Gussekloo et al., 2017*). However, our understanding of the transformation of the palate from the ancestral heavily built and immobile dinosaurian morphology into the flexible derived crown bird condition has been hindered by a lack of detailed early avialan skull records (*Witmer and Martin, 1987*; *Field et al., 2018b*; *Hu et al., 2019*). Reconstruction of the complete palate of *Yuanchuavis* reveals a stout vomer that is dorsoventrally compressed and likely forms a large overlapping contact with the maxilla/premaxilla as in *Sapeornis* and non-avialan theropods (*Figure 3*, *Figure 3—figure supplement 3*; *Ostrom, 1969*; *Holliday and Witmer, 2008*). The pterygoid is morphologically nearly identical to that of non-avialan theropods in having a dorsally oriented quadrate ramus that extends medially bypassing the orbital process of the quadrate, resulting in a scarf joint widely distributed among non-avialan dinosaurs (*Holliday and Witmer, 2008*; *Tsuihiji et al., 2014*). The forked condition of this quadrate ramus is shared with close relatives of Avialae like *Linheraptor*, and this dinosaurian pterygoid previously has been recognized only among early avialans in the juvenile enantiornithine IVPP V12707 (*Wang et al., 2021a*). The presence of this plesiomorphic morphology in both a juvenile (i.e. IVPP V12707) and a skeletally mature (here, *Yuanchuavis*) enantiornithine eliminates ontogenetic variation as a potential source for its morphology and signifies conclusive evidence as to the presence of the kind of dinosaurian pterygoid in enantiornithines, along with its functional implications.

The palatine exhibits a peculiar morphology distinguishable from that of other early-diverging avialans and non-avialan theropods in having a medially directed choanal process, having a caudo-medially oriented pterygoid process, and lacking a jugal process. To explore evolutionary changes in palatine morphology across the theropod to bird transition, we performed landmark-based geometric morphometric analyses. A phylomorphospace was created by using the first three principal components (PCs 1–3: >90% of shape variances) calculated from generalized Procrustes superimposition and followed by phylogenetic principal components analysis (pPCA) to account for phylogenetic non-independence (*Figure 3h and i*; *Figure 3—figure supplement 4*). PC1 mainly describes the slenderness of the palatine, with *Yuanchuavis* and the crown bird *Dromaius*, exhibiting the lowest score (mediolaterally more slender; *Figure 3h*). PC2 corresponds to the length of the jugal process and the overall shape of the palatine; differences along this axis distinguish most avialans from non-avialan theropods. PC3 is mainly aligned with the rostrocaudal length of the palatine, and *Yuanchuavis* and *Archaeopteryx* are clustered with most other non-avialan theropods. *Yuanchuavis* is widely separated from other stem and crown avialans as well as representatives of major non-avialan theropods in palatine morphospace (*Figure 3h and i*; *Figure 3—figure supplement 4*). Presumably, the loss of the jugal process represents the major structural change of the palatine in early bird evolution. The release of constraints imposed by contacts with other elements, combined with selection for derived feeding behaviors and functions, likely guided the morphological and functional divergence of the palatine. Future studies with increased species sampling are needed to further test this hypothesis.

With >10,000 species, neognaths are the most morphologically and ecologically diverse clade of modern amniotes (*Gill, 2007*), with much of their success attributed to key evolutionary novelties such as powered flight, but also their uniquely kinetic crania (*Zusi, 1993*; *Gussekloo and Bout, 2005*; *Toews et al., 2016*). Avian cranial kinesis has been demonstrated to have improved feeding performance by increasing biting force, jaw closing speed, and food handling precision (*Bock, 1964*; *Zusi, 1984*). Avian cranial kinesis works via pathways composed of two kinetically permissive linkage systems: the quadrate-quadratojugal-jugal-rostrum on the lateral margin and the quadrate-pterygoid-palatine-vomer on the palatal aspect, which coordinate in transmitting force and movement of the musculature to the rostrum through the sliding movement of the entire palate (*Figure 4*; *Bock, 1964*; *Holliday and Witmer, 2008*; *Bhullar et al., 2016*; *Wang et al., 2021a*). However, to achieve this movement, the involved elements have been radically transformed from the primitive condition in non-avialan theropods, in which these cranial bones are robust and articulate with each other through immobile and largely sutural contacts, to lighter elements with condylar or otherwise reduced contacts (*Figure 4a, b*; *Holliday and Witmer, 2008*). The CT reconstruction demonstrates that neither of these two pathways utilized in modern cranial kinesis are present or functional in *Yuanchuavis*. The rostro-caudal movement of the jugal bar to protract/retract the upper jaw is prevented by the complete temporal bars and the bracing of the ectopterygoid. The sliding movement of the palatal elements is severely restricted by the prominent basipterygoid process (that fits into a cotyle on the pterygoid) and the lack of permissive synovial articulations at the quadrate-pterygoid and pterygoid-palatine contacts (*Figure 4c, d*).

However, as noted previously (*Wang et al., 2021a*), the presence of the scarf joint between the quadrate and the pterygoid along with the potential rotation of the quadrate mediolaterally around its long dorsoventral axis as in crown birds (*Zusi, 1984*) may have allowed for some rudimentary force transmission from the quadrate rostrally along the palatal bone contacts with the basipterygoid process acting as a fulcrum. That hypothesized movement among early avialans may have been the functional foundation from which other features were evolved and exapted to complete the mosaic of modern avian cranial kinesis. Our study confirms that the akinetic skull present in earliest-diverging avialans was retained in enantiornithines (*Wang et al., 2021a*). Despite their akinetic origins, features such as the loss of the jugal process of the palatine and further reduction in the bones of the temporal region would go on to be exapted and modified further as part of the evolutionary assembly of kinetic skulls. Many of the skeletal modifications necessary for cranial kinesis were pieced together along the stem of the extant sister group of the Enantiornithes (*Witmer and Martin, 1987*; *Wang et al., 2021a*), with the oldest known records of the neognath style of palatine-pterygoid contact, the condylar contact between the quadrate and pterygoid, and loss of the diapsid temporal condition in the Late Cretaceous *Ichthyornis* and *Hesperornis* (*Elżanowski, 1991*; *Field et al., 2018b*).

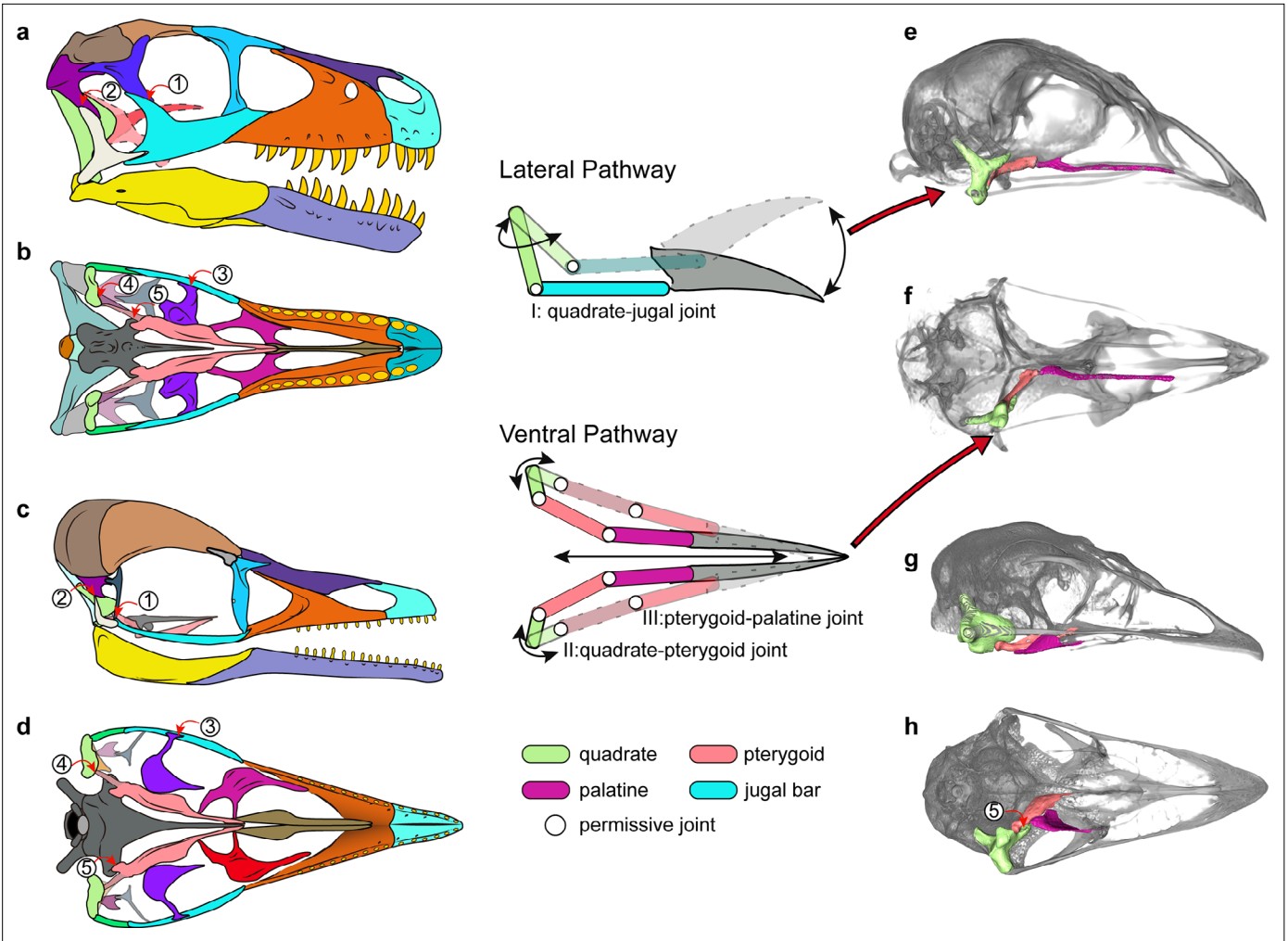

**Figure 4.** Cranial kinesis in early bird evolution. The typical avian cranial kinesis is realized through two pathways (schematic drawing in the middle): the quadrate-jugal bar-rostrum on the lateral side and quadrate-pterygoid-palatine on the ventral side, highlighted in the galliform *Tragopan caboti* (**e** and **f**). These two pathways are restricted by the postorbital bar (1), squamosal-quadratojugal contact (2), ectopterygoid (3), the scarf joint between quadrate-pterygoid (4), and the prominent pointed basipterygoid process (5) in non-avialan theropods (**a** and **b**) and enantiornithines (**c** and **d**), indicating an akinetic skull. The ventral pathway is also absent in most paleognaths (**h**) at least restricted by non-permissive contact (5). (**a–d**) Skull reconstructions of *Dromaeosaurus* (Theropod: Dromaeosauridae) in lateral (**a**) and ventral (**b**) view (modified from *Currie, 1995*), *Yuanchuavis* (Avialae: Enantiornithes) in lateral (**c**) and ventral (**d**) view. (**e–g**) Digital renderings of skulls of *Tragopan caboti* (Galloanserae: Galliformes) in lateral (**e**) and ventral (**f**) view, and *Dromaius novaehollandiae* (Palaeognathae: Casuariiformes) in lateral (**g**) and ventral (**h**) view.

The mixture of plesiomorphic temporal and palatal regions and derived facial anatomies captured in this single enantiornithine skull vividly demonstrates how the avialan cranium has been shaped deeply by evolutionary mosaicism—a hypothesis proposed by studies involving crown birds but still awaiting paleontological evidence (*Bhullar et al., 2016*; *Felice et al., 2020*; *O'Connor et al., 2020*; *Plateau and Foth, 2020*). That difference is even more conspicuous between the skull and the post-cranial skeleton of *Yuanchuavis*, given the presence of derived avialan features such as heterocoelic cervical vertebrae (*Figure 1—figure supplement 2*), a fused synsacrum and pygostyle, and a reversed hallux (*Wang et al., 2021b*), demonstrating that the avialan bauplan is highly modular (*Orkney et al., 2021*). The Enantiornithes were the dominant avialan clade over a large part of the Cretaceous around the globe with their enhanced locomotion (powered flight) and diverse feeding strategies conveyed by disparate rostrum morphologies, hyoid, and dental morphologies (*O'Connor and Chiappe, 2011a*; *O'Connor et al., 2011b*; *O'Connor et al., 2020*; *Wang et al., 2021b*), suggesting that the relationship between a kinetic skull and diversification is context dependent and that the timing of such link likely arose after the divergence of the Enantiornithes. Comparing the extinction of enantiornithines

at the end of the Cretaceous to the survival of members of their sister clade Ornithuromorpha (*Field et al., 2018a*; following *O'Connor et al., 2016b*, the Ornithuromorpha refers to Neornithes and species that are more closely related to it than to the Enantiornithes), it is easy to hypothesize that the ancestral akinetic avialan skull, despite its successful use among avialans throughout the Cretaceous, did not allow for the adaptations or extreme selection afforded by the fully kinetic skulls of crown birds. That kinesis with its origin spread across various points of the Mesozoic avialan tree has contributed to the greatest radiation of avialans and their occupation of diverse habitats around the world.

## Methods

### Revised systematic paleontology of *Y. kompsosoura*

Avialae *Gauthier, 1986*.
Ornithothoraces *Chiappe, 1995*.
Enantiornithes *Walker, 1981*.
Pengornithidae *Wang et al., 2014*.
*Y. kompsosoura Wang et al., 2021b*.

### Revised diagnosis

A large pengornithid enantiornithine that is distinguishable from other pengornithids on the basis of the following features (*autapomorphy): premaxilla bearing six teeth with its rostral tip edentulous*; lacrimal bearing a lateral flange; dorsoventrally broad orbital process of the quadrate; unfused vomers bearing a transverse dorsal ridge*; palatine having a medially directed choanal process and lacking a jugal process; pterygoid with a forked quadrate ramus; cranial thoracic vertebrae having well-developed hypapophyses; middle thoracic centra laterally excavated by broad fossae; and pygostyle with elongate ventrolateral processes and short dorsal processes (modified from *Wang et al., 2021b*).

### X-ray CT imaging

To optimize the scanning resolution, the skull and the two cranial vertebrae (atlas and axis) of the holotype of *Y. kompsosoura* (IVPP V27883) were isolated from the slab. The skull was scanned using the industrial CT scanner Phoenix v-tome-x at the Key Laboratory of Vertebrate Evolution and Human Origins, Institute of Vertebrate Paleontology and Paleoanthropology (IVPP, Beijing, China), with a beam energy of 140 kV and a flux of 150 μA at a resolution of 11.68 μm per pixel. The resulting scanned images were imported into Avizo (version 9.2.0) for digital segmentation, rendering, and reconstruction. The obtained three-dimensional models were optimized in MeshLab (version 2012.12). CT data of the enantiornithine IVPP V12707 and two modern birds (*Tragopan caboti* and *Dromaius novaehollandiae*) produced during our previous research (*Wang et al., 2021a*) were re-analyzed and rendered for comparison.

### Geometric morphometric analysis

In order to trace changes of palatine morphology among early-diverging avialans and their close non-avialan theropod relatives, a landmark-based geometric morphometric analysis was performed. The palatine is generally poorly preserved in early avialan and non-avialan theropods, and we only have been able to assemble a dataset of 14 fossil taxa that preserve this element intact, including nine non-avialan theropods and five Mesozoic avialans (*Supplementary file 1*). Two crown birds—*Dromaius novaehollandiae* (Palaeognathae: Casuariiformes) and *Tragopan caboti* (Neognathae: Galliformes)—were included for comparison. Despite the small sample size, the dataset contains species from the major clades of non-avialan theropods and early avialans and, therefore, should encompass the morphological disparity of palatine with moderate sufficiency. 17 landmarks and 1 curve (15 evenly spaced semi-landmarks along the lateral margin of the palatine) were digitized on the two-dimensional geometry of the palatine in dorsal/ventral aspect using the tpsDIG software (see *Figure 3—figure supplement 5* and Appendix 1 for landmark schemes; *Rohlf, 2009*). A three-dimensional geometric morphometric analysis was not employed here because this method would reduce the sample size greatly and add little additional information considering that the palatine is a dorsoventral sheet-like element in most taxa. To remove the non-biological effects of rotation, scaling, and translation

(*Zelditch et al., 2012*), the raw coordinate data was transformed by generalized Procrustes analysis (GPA), with the semi-landmarks slid to minimal bending energy, using the gpagen function in the R package geomorph (version 4.0.0; *Adams et al., 2013*). To account for the non-independence of phenotypes among taxa because of shared history, a pPCA was performed (*Revell, 2009*). First, a supertree containing only the focal taxa was compiled with reference to recent phylogenetic work (*Turner et al., 2012*; *Wang et al., 2021b*). This phylogeny was time calibrated using tip dates bracketed by the first and last appearance datum of the geological stages or epochs in which each taxon was collected (*Brusatte, 2011*). Zero-length branches were smoothed using the 'minimum branch length' embedded in the timePaleoPhy function in the R package paleotree (*Bapst, 2012*). Then, the GPA-transformed coordinate data and time-calibrated phylogeny were subjected to pPCA using the gm.prcomp function in the R package geomorph, which performed a generalized least square estimation of a covariance matrix (*Collyer et al., 2021*). A phylomorphospace was constructed using the principal component scores of taxa and ancestral nodes to visualize changes of palatine geometry along the line to early avialans (*Figure 3h, i*).

In addition, a traditional geometric morphometric analysis also was applied for comparison (*Figure 3—figure supplement 4*). The first three PCs account for 82.13% of the shape variance. PC1 corresponds to the discrepancy between the medial distance defined by the choanal and pterygoid processes and the lateral distance defined by the maxillary and jugal processes, with *Yuanchuavis* exhibiting the lowest scores. PC2 corresponds to the transverse width of the palatine body, and differences along this axis distinguish all avialans except *Archaeopteryx* from non-avialan theropods. PC3 is most aligned with the relative elongation of the maxillary process and the shortness of the pterygoid process, with *Yuanchuavis* and *Archaeopteryx* separated from other avialans.

## Acknowledgements

We thank Song Miao and Jiutong Feng for helping in CT data scanning and segmentation. We thank Han Hu for sharing photographs of *Sapeornis*. This research is supported by the Key Research Program of Frontier Sciences, CAS (ZDBS-LY-DQC002), the National Natural Science Foundation of China (42288201, 42172029), and the Tencent Foundation (through the XPLORER PRIZE).

## Additional information

### Funding

| Funder | Grant reference number | Author |
| --- | --- | --- |
| Key Research Program of Frontier Science, Chinese Academy of Sciences | ZDBS-LY-DQC002 | Min Wang |
| National Natural Science Foundation of China | 42288201 | Zhonghe Zhou |
| National Natural Science Foundation of China | 42172029 | Thomas A Stidham |
| Tencent | XPLORER PRIZE | Min Wang |

The funders had no role in study design, data collection and interpretation, or the decision to submit the work for publication.

### Author contributions

Min Wang, Conceptualization, Data curation, Formal analysis, Supervision, Funding acquisition, Validation, Investigation, Visualization, Methodology, Writing – original draft, Project administration, Writing - review and editing; Thomas A Stidham, Jingmai K O'Connor, Formal analysis, Investigation, Writing – original draft; Zhonghe Zhou, Formal analysis, Supervision, Funding acquisition, Investigation, Writing – original draft

### Author ORCIDs

Min Wang (iD) http://orcid.org/0000-0001-8506-1213

Decision letter and Author response
Decision letter https://doi.org/10.7554/eLife.81337.sa1
Author response https://doi.org/10.7554/eLife.81337.sa2

## Additional files

### Supplementary files
- MDAR checklist
- Supplementary file 1. Taxa used in geometric morphometric analysis of palatine shape.

### Data availability

The specimen (IVPP V27883) described in this study is archived and available on request from the Institute of Vertebrate Paleontology and Paleoanthropology (IVPP), Chinese Academy of Sciences, Beijing, China. Additional figures of cranial anatomy are available in the Supplementary Information. The three-dimensional models (STL) are archived and available on Dryad, or from the corresponding authors.

The following dataset was generated:

| Author(s) | Year | Dataset title | Dataset URL | Database and Identifier |
|---|---|---|---|---|
| Wang M, Stidham T, O'Connor J, Zhou Z | 2022 | CT data for Insight into the evolutionary assemblage of cranial kinesis from a Cretaceous bird | https://dx.doi.org/10.5061/dryad.bnzs7h4dq | Dryad Digital Repository, 10.5061/dryad.bnzs7h4dq |

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

# Appendix 1

## Anatomical description of landmarks and semi-landmarks positions

Landmarks 1–17 (*Figure 3—figure supplement 5*):

1. Rostral tip of the maxillary process.
2. Medial midpoint of the maxillary process.
3. Medial deflection point between the maxillary process and palatine body.
4. Midpoint of medial margin connecting the maxillary and choanal processes.
5. Rostral deflection point between the choanal process and palatine body.
6. Rostral midpoint of the choanal process.
7. Rostral tip of the choanal process.
8. Caudal midpoint of the choanal process.
9. Caudal deflection point between the choanal process and palatine body.
10. Midpoint of medial margin connecting the choanal and pterygoid processes.
11. Rostral deflection point between the pterygoid process and palatine body.
12. Rostral midpoint of the pterygoid process.
13. Caudal tip of the pterygoid process.
14. Caudal midpoint of the pterygoid process.
15. Caudal deflection point between the pterygoid process and palatine body.
16. Lateral deflection point between the maxillary process and palatine body.
17. Lateral midpoint of the maxillary process.

Semi-landmarks: 15 semi-landmarks are placed along the curve connecting landmarks 15 and 16.

