## [Editor Report]

Most birds today can lift the upper beak independently of the brain case, enabled by a series of mobile joints and bending zones in the skull. The computed tomography of the skull of a 120-million-year-old toothed bird produced by the authors shows for the first time that the joints were still absent, but also hints at how they may have evolved later. This compelling, important paper is of high interest to evolutionary biologists, vertebrate paleontologists (especially, but by no means only, those working on bird origins) and specialists in biomechanics.

---

## [Decision Letter]

**Decision letter after peer review:**

Thank you for submitting your article "Insight into the evolutionary assemblage of cranial kinesis from a Cretaceous bird" for consideration by *eLife*. Your article has been reviewed by two peer reviewers, including David Marjanović as the Reviewing Editor and Reviewer #1, and the evaluation has been overseen by George Perry as the Senior Editor. The following individuals involved in the review of your submission have agreed to reveal their identity: by Christopher Torres (Reviewer #2).

I wrote my review before reading the one by Dr. Torres, so please consider and compare both reviews carefully. Being more familiar than I am with the pertinent literature, Dr. Torres raised an important point that I overlooked: the palatines of *Ichthyornis* were misidentified in Field et al. (2018), and the real palatines were identified in Torres et al. (2021). The other major point of substance is the uncited paper by Elżanowski et al. (2018, cited below) on the skull of *Confuciusornis* and the lack of a citation for the skull of *Sapeornis*. The other comments in both reviews concern clarity of presentation, broadly speaking.

*Reviewer #1 (Recommendations for the authors):*

I recommend publication in *eLife* after minimal revision.

Lines 342-343: The condition in confuciusornithiforms is much weirder and may not be derived directly from the diapsid condition (Elżanowski et al., 2018). As far as I understand, this reference should be cited instead of Chiappe et al. (1999) and Hou et al. (1999). Given that you are no doubt aware of this reference, perhaps you have reasons not to cite it; in that case, however, those reasons should be spelled out. In any case, a reference for Sapeornis should be added – you don't currently have one in that sentence.

Elżanowski A, Peters DS, Mayr G. 2018. Cranial morphology of the Early Cretaceous bird Confuciusornis. Journal of Vertebrate Paleontology 38(2):e1439832. DOI: https://doi.org/10.1080/02724634.2018.1439832

Comments on terminology and nomenclature

147, 149, 204, 255 (twice), 256, 258, 261 (twice), 264, 265, 285, 341, 441, 601, 603, 606, Supplementary File 2 under Hesperornis and in the references list: Elżanowski with ż. The sound this letter represents is approximately that found in English words like 'vision' or 'Asia' and identical to some versions of a Chinese r.

156: "paroccipital"; Greek and (especially) Latin have an aversion to vowel clusters – compare 'autapomorphy' or 'hypapophysis' (auto-, hypo-).

437: Ornithuromorpha isn't crownward, it contains the crown. (And should be Euornithes Sereno, 1998, anyway.) I suggest "along the stem of the extant sister-group of Enantiornithes".

459: At least two definitions of the name Ornithuromorpha make that clade a bit smaller than the sister clade of Enantiornithes. Euornithes does not have this problem…

584: "Brusatte SL", "Jarvis ED".

614, 717: italics

646: In 2012, the PLOS journals changed from "PLoS" to "PLOS", announced that decision on their blog, changed their logos and everything – everything except the "suggested citation" in each paper. Having published in PLOS ONE myself, I wrote to them back in 2015 about this; I never got a reaction. I recommend citing the journals as "PLOS" for papers published in 2012 or later.

667 and in the references to Suppl. File 2: "Madsen JH Jr" – he's the younger James H., not the younger of only two Madsens.

741, 761: "PalAsiatica" (possibly the oldest CamelCase in the world).

749: Géologie, Faculté.

References to Suppl. File 2: Italics for Ichthyornis, uppercase for Alioramus!

*Reviewer #2 (Recommendations for the authors):*

Line 50-54 – I urge citing previous authors who have hypothesized this connection between a kinetic skull and the rapid avian radiation.

Line 58-62 – The use of "overzealously" reads as unnecessarily editorial. Also, this statement is quite vague – what "certain bony contacts" are meant here?

Line 334 – Elsewhere, the authors described the shape of the postorbital as "y-shaped," which I find a more apt description.

Figure 2 – I recommend clarifying in the caption that panel b is a cut-away of a (unless I'm wrong?).

Figure 3 – Panels a and b are described as "lateral" and "ventral" view, but I don't think that's right – they appear to be opposing views.

Figure 4 – Again, I generally find this to be a superlative figure. My only major issue is the 3D models on the right side. I recommend recoloring the bones that are not of interest a neutral gray, so that they match each other and so that the bones that are of interest are better highlighted (the red base of the skull in g-h is also distracting). I also recommend recoloring the bones of interest to better match the cartoons and the schematic models. "Pathway" is misspelled.

Supplemental Figure 3-2 – This figure is very useful, but that usefulness is somewhat negated by the inconsistency of orientations and anatomical views of the models and cartoons. I strongly recommend that all panels b-f be rotated to match panels a and g-h. I also strongly recommend all the elements be shown in the same anatomical view (either ventral or dorsal).

Supplemental Figure 3-3 – I recommend that either panel a or b be mirrored so as to optimize their comparability.

[Editors' note: further revisions were suggested prior to acceptance, as described below.]

Thank you for resubmitting your work entitled "Insight into the evolutionary assemblage of cranial kinesis from a Cretaceous bird" for further consideration by *eLife*. Your revised article has been evaluated by George Perry (Senior Editor) and a Reviewing Editor.

The manuscript is almost ready for publication! Both reviews are very short and only have stylistic comments anymore:

*Reviewer #1 (Recommendations for the authors):*

The manuscript has improved further; all requested changes seem to have been made.

418-419: It took me surprisingly long to parse this sentence. I recommend: "[…] palatine; differences along this axis distinguish most avialans […]".

420: Replace the second comma with another "and".

430: Replace ", and" by "as well as".

476: "the earliest-diverging".

503: After "sister clade" would be a good place to cite your preferred definition of Ornithuromorpha.

*Reviewer #2 (Recommendations for the authors):*

The revised manuscript addresses all of my major concerns.

---

## [Author Response]

Reviewer #1 (Recommendations for the authors):I recommend publication in eLife after minimal revision.Lines 342-343: The condition in confuciusornithiforms is much weirder and may not be derived directly from the diapsid condition (Elżanowski et al., 2018). As far as I understand, this reference should be cited instead of Chiappe et al. (1999) and Hou et al. (1999). Given that you are no doubt aware of this reference, perhaps you have reasons not to cite it; in that case, however, those reasons should be spelled out. In any case, a reference for Sapeornis should be added – you don't currently have one in that sentence.Elżanowski A, Peters DS, Mayr G. 2018. Cranial morphology of the Early Cretaceous bird Confuciusornis. Journal of Vertebrate Paleontology 38(2):e1439832. DOI: https://doi.org/10.1080/02724634.2018.1439832

We thank you for bringing out this important reference, and have cited Elżanowski et al. (2018), and Hu et al. (2019) for *Confuciusornis* and *Sapeornis*, respectively.

Comments on terminology and nomenclature147, 149, 204, 255 (twice), 256, 258, 261 (twice), 264, 265, 285, 341, 441, 601, 603, 606, Supplementary File 2 under Hesperornis and in the references list: Elżanowski with ż. The sound this letter represents is approximately that found in English words like 'vision' or 'Asia' and identical to some versions of a Chinese r.

Yes, we have revised the error throughout the manuscript.

156: "paroccipital"; Greek and (especially) Latin have an aversion to vowel clusters – compare 'autapomorphy' or 'hypapophysis' (auto-, hypo-).

Yes, these mistakes have been corrected.

437: Ornithuromorpha isn't crownward, it contains the crown. (And should be Euornithes Sereno, 1998, anyway.) I suggest "along the stem of the extant sister-group of Enantiornithes".

Yes, we have revised this sentence accordingly.

459: At least two definitions of the name Ornithuromorpha make that clade a bit smaller than the sister clade of Enantiornithes. Euornithes does not have this problem…

The term Ornithuromorpha Chiappe 2002, was originally a node-based definition, referring to the clade including the common ancestor of *Patagopteryx* and Ornithurae plus all its descendants. This definition was brought out when sparce Mesozoic ornithuromorphs were known. Since then, a handful of Mesozoic birds have been discovered and reported, and our understanding of phylogeny of Mesozoic birds has been greatly refined by these fossils. In most recent studies, it is widely accepted that the Ornithuromorpha is the sister clade to the Enantiornithes, rather than more exclusively only referring to the clade uniting *Patagopteryx* and more crownward taxa. Even the recent studies of Dr. Chiappe also share this understanding and labeled “Ornithuromorpha” as the sister clade of the Enantiornithes. To clarify this issue, we adopt the revised definition of Ornithuromorpha by O’Connor et al. 2016, which formally defined the Ornithuromorpha as “the first ancestor of Neornithes that is not also an ancestor of the Enantiornithes, and all of its descendants”. We have revised this content in the manuscript.

O’Connor, J. K., M. Wang, and H. Hu. 2016. A new ornithuromorph (Aves) with an elongate rostrum from the Jehol Biota, and the early evolution of rostralization in birds. Journal of Systematic Palaeontology 14:939-948.

584: "Brusatte SL", "Jarvis ED".614, 717: italics646: In 2012, the PLOS journals changed from "PLoS" to "PLOS", announced that decision on their blog, changed their logos and everything – everything except the "suggested citation" in each paper. Having published in PLOS ONE myself, I wrote to them back in 2015 about this; I never got a reaction. I recommend citing the journals as "PLOS" for papers published in 2012 or later.667 and in the references to Suppl. File 2: "Madsen JH Jr" – he's the younger James H., not the younger of only two Madsens.741, 761: "PalAsiatica" (possibly the oldest CamelCase in the world).749: Géologie, Faculté.References to Suppl. File 2: Italics for Ichthyornis, uppercase for Alioramus!

We thank Dr. Marjanović for these detailed comments and have corrected all these mistakes accordingly.

Reviewer #2 (Recommendations for the authors):Line 50-54 – I urge citing previous authors who have hypothesized this connection between a kinetic skull and the rapid avian radiation.

Yes, we have added relevant references here.

Line 58-62 – The use of "overzealously" reads as unnecessarily editorial. Also, this statement is quite vague – what "certain bony contacts" are meant here?

Yes, we have removed the word “overzealously”.

Line 334 – Elsewhere, the authors described the shape of the postorbital as "y-shaped," which I find a more apt description.

We removed the word “T”, and instead said “free postorbital”.

Figure 2 – I recommend clarifying in the caption that panel b is a cut-away of a (unless I'm wrong?).

Panel a and b both refer the right premaxilla. Panel a is right lateral view. Panel b is the same element that is mirrored to the medial view. We have explained this in the caption.

Figure 3 – Panels a and b are described as "lateral" and "ventral" view, but I don't think that's right – they appear to be opposing views.

Panel a and b both refer the right pterygoid in lateral and medial view, respectively. We took the screenshot of the medial view in Avizo and then flipped the image, to make it aligned with Panel a with the rostral end on the right side.

Figure 4 – Again, I generally find this to be a superlative figure. My only major issue is the 3D models on the right side. I recommend recoloring the bones that are not of interest a neutral gray, so that they match each other and so that the bones that are of interest are better highlighted (the red base of the skull in g-h is also distracting). I also recommend recoloring the bones of interest to better match the cartoons and the schematic models. "Pathway" is misspelled.

Yes, we took the advice and redesigned Figure 4. All the cranial elements in the 3D model that are not focus in this study are colored in grey. We also recolored the bones of interest among the line drawings and 3D models.

Supplemental Figure 3-2 – This figure is very useful, but that usefulness is somewhat negated by the inconsistency of orientations and anatomical views of the models and cartoons. I strongly recommend that all panels b-f be rotated to match panels a and g-h. I also strongly recommend all the elements be shown in the same anatomical view (either ventral or dorsal).

Yes, we have revised this figure according to the comment.

Supplemental Figure 3-3 – I recommend that either panel a or b be mirrored so as to optimize their comparability.

Yes, this figure has been revised accordingly.

[Editors' note: further revisions were suggested prior to acceptance, as described below.]

Thank you for resubmitting your work entitled "Insight into the evolutionary assemblage of cranial kinesis from a Cretaceous bird" for further consideration by eLife. Your revised article has been evaluated by George Perry (Senior Editor) and a Reviewing Editor.The manuscript is almost ready for publication! Both reviews are very short and only have stylistic comments anymore:Reviewer #1 (Recommendations for the authors):The manuscript has improved further; all requested changes seem to have been made.418-419: It took me surprisingly long to parse this sentence. I recommend: "[…] palatine; differences along this axis distinguish most avialans […]".420: Replace the second comma with another "and".430: Replace ", and" by "as well as".476: "the earliest-diverging".503: After "sister clade" would be a good place to cite your preferred definition of Ornithuromorpha.

Yes, we have revised all these paragraphs accordingly, and the phylogenetic definition of the Ornithuromorpha used in present study is specified in the corresponding paragraph.